# Peculiarities in the Amino Acid Composition of Sow Colostrum and Milk, and Their Potential Relevance to Piglet Development

**DOI:** 10.3390/vetsci10040298

**Published:** 2023-04-17

**Authors:** Renjie Yao, An Cools, Anneleen Matthijs, Peter P. De Deyn, Dominiek Maes, Geert P. J. Janssens

**Affiliations:** 1Department of Veterinary and Biosciences, Faculty of Veterinary Medicine, Ghent University, 9820 Merelbeke, Belgium; 2Department of Internal Medicine, Reproduction and Population Medicine, Faculty of Veterinary Medicine, Ghent University, 9820 Merelbeke, Belgium; 3Laboratory of Neurochemistry and Behaviour, Faculty of Pharmaceutical, Biomedical and Veterinary Sciences, Faculty of Medicine and Health Sciences, University of Antwerp, 2610 Antwerp, Belgium

**Keywords:** amino acids, colostrum, sow milk, creep feed

## Abstract

**Simple Summary:**

Sow’s milk, serving as the primary nutrient source, contributes to the morphological and functional maturation of piglets’ gastrointestinal tract. The abrupt diet transition at weaning has required piglets to familiarize a solid diet earlier through supplementation with creep feed apart from milk during the suckling period. We assume that the composition of sow milk is the gold standard and could inspire the optimization of creep feed for suckling piglets. Therefore, in this study, we investigated how conserved and variable the amino acid profile of sow colostrum and milk is, compared with other literature and species. The amino acid profile of sow milk is conserved compared to actual content throughout the lactation period and is rich in glycine, valine, and proline compared to other species. Similarities with characteristic differences were observed within these comparisons, which may reflect the nutritional requirements for preweaning piglets.

**Abstract:**

The composition of mother’s milk is considered the ideal diet for neonates. This study investigated how conserved or variable the amino acid profile of sow colostrum and milk is throughout lactation, compared with other studies in sows and other species. Twenty-five sows (parity one to seven) from one farm with gestation lengths of 114 to 116 d were sampled on d 0, 3, and 10 after parturition. The total amino acid profile of the samples was analyzed through ion-exchange chromatography, and the results were displayed as the percentage of total amino acid and compared with literature data. Most of the amino acid concentrations in sow milk decreased significantly (*p <* 0.05) throughout the lactation period, while the amino acid profile generally showed a conserved pattern, especially from d 3 to d 10, and was rather similar across different studies. Glutamine + glutamate was the most abundant amino acid in milk at all sampling moments, accounting for 14–17% of total amino acids. The proportions of proline, valine, and glycine in sow milk nearly accounted for 11%, 7%, and 6% respectively, and were higher compared to human, cow, and goat milk, while the methionine proportion was less than the other three. Compared to the large variations often reported in macronutrient concentrations, the amino acid profile of sow milk in the present study, as well as in others, seems well conserved across the lactation period. Similarities with characteristic differences were also observed between sow milk and piglet body composition, which might reflect the nutrition requirements of preweaning piglets. This study warrants further research exploring the link between the whole amino acid profile and the particular amino acids for suckling piglets and could facilitate insight for optimizing creep feed.

## 1. Introduction

Amino acids from sow milk play a key role in offspring development. Apart from their traditional role as “building blocks” for protein synthesis, amino acids also have important functional roles in metabolism [1]. They are involved in features such as regulating gene expression and cell signaling [2,3,4]. They participate in the maintenance, growth, and immunity of organisms. For example, proline is dominantly present in collagen and is also involved in the activation of the mammalian target of rapamycin (mTOR) cell signaling for initiating protein synthesis [5,6,7]. Amino acids can be used as alternative sources of other important metabolites, such as glucose, fatty acids, and nitric oxide. Specifically, branched-chain amino acids (BCAA) can be catabolized in the mammary gland to branched-chain α-keto acids (BCKA), which can be further decarboxylated into a series of metabolites during lactation which enter the citric acid cycle for producing energy in the mammary gland [8,9,10,11]. These examples demonstrate that amino acids are not only necessary as structural components of proteins, they can also support the gut development of neonatal piglets.

Amino acids are traditionally classified as essential amino acids (EAA) and nonessential amino acids (NEAA). The EAAs are those that the organisms cannot synthesize de novo and, therefore, they must be supplied by the diet [12]. This definition may lead to an underestimation of the functions and importance of some NEAA. Therefore, some NEAA are reconsidered as conditionally essential amino acids that the organisms normally can synthesize in adequate amounts, but which must be obtained from the diet on particular occasions [13]. For instance, the NEAA glutamine and glutamate, the two most abundant amino acids in sow milk, are synthesized by the mammary tissue starting from BCAA and subsequently secreted into the milk. This means that the uptake of BCAA by the sow’s mammary gland is higher than their output in milk [14,15]. This conversion of amino acids in the mammary gland during lactation is reasonably considered a natural process to optimize the diet of the offspring. These observations suggest that these abundant amino acids in milk may have a high requirement in the newborn and that supplementation of these NEAA may be beneficial for the intestinal health and growth of suckling piglets [16,17,18,19,20]. From an evolutionary perspective, the amino acid profile in milk should be used as the gold standard for understanding the nutritional requirements of neonates [21]. Many amino acids display remarkable metabolic versatility, and it can be hypothesized that natural evolution has optimized the amino acid composition of sow colostrum and milk to support neonatal piglet growth. 

The present study investigated the amino acid profile of sow colostrum and milk. Knowledge about the amino acid profile and comparison with the body composition of suckling piglets and other species’ amino acid profile of milk may contribute to the optimization of feeding suckling piglets. 

## 2. Materials and Methods

### 2.1. Animals and Housing

In the present experiment, 25 sows (breed: Topigs 20; with average parity of 3.65 ± 1.65 (min 1–max 7)) were selected for colostrum and milk sampling. The sows had a gestation length of 114 to 116 days and were housed in a commercial farm in Belgium. The day of farrowing was defined as d 0 of lactation. During the whole lactation period, all sampled sows were managed in the same way according to the regular procedures of the commercial farm. The average litter size of these sows was 15.5, with, on average, 13.3 live-born piglets. The sows were offered the same lactation diet and the feeding scheme was identical for all sows. All diets applied for sows in this commercial farm met the nutrient requirements for this pig category and feeding stage. They were housed in a climate-controlled farrowing unit with a 14 h light and 10 h dark cycle. Each sow had access to an individual drinking nipple and water was available unrestrictedly with a nipple flow rate of 1.5–2 L/min. The health condition of the sows on this farm was supervised continuously. 

### 2.2. Milk Samples and Chemical Analysis

The colostrum and milk (around 25 mL per sow) were collected from all teats along one side of the mammary glands within 10 h after the birth of the first piglet (d 0) and at d 3 and d 10 of lactation. At each sampling point, the milk samples were collected 10 min after intramuscular injection of 2 mL oxytocin (10 IU/mL) in the necks of sows. Samples from the different teats of each sow were pooled immediately upon collection and stored at −20 °C until further analysis.

The total amino acid composition of all of the samples was determined after hydrolysis of the sample. The samples were first dried with a Savant speed-vac system, after which they were further dried in a desiccator with potassium hydroxide platelets and phosphorus pentoxide. Then, 6 N HCl containing 1% phenol and 6% thioglycolic acid was added to the dried samples. Subsequently, the hydrolysis was carried out under a vacuum and inert conditions with nitrogen gas to prevent oxidative degradation of the amino acids during the hydrolysis. The samples were heated at 110 °C for 24 h and then dried under a vacuum. The samples were washed several times with a solution of water, ethanol, and triethylamine (2:2:1 *v*/*v*) to remove any acid residue. Lithium citrate buffer was added to the dry hydrolysis product and dilutions were made for amino acid analysis with a Biotronik LC 6001 Amino Acid Analyzer (Biotronik, Maintal, Germany). The optical density was measured by the ninhydrin method. As a result of sample hydrolyses, no distinction between glutamine and glutamate and between asparagine and aspartate was possible. Therefore, these amino acids are always reported as glutamine + glutamate and asparagine + aspartate. 

### 2.3. Data Selection

Data on amino acid profiles in whole piglet body composition and cow, goat, and human milk were collected from published studies [22,23,24,25,26,27,28,29,30,31]. Cow and goat milk nearly account for nearly 85% of the global production of milk and dairy products for human consumption, and humans share similar metabolic and digestive processes with pigs [32,33,34]. Therefore, we have made a comparison among these species. We searched in PubMed and Web of Science using the keywords “amino acid composition”, “sow milk”, “breast milk” and “piglets body composition” in September 2021. Our selection criteria for the literature search were: (1) The types of amino acids which were analyzed should include all those we had analyzed, (2) The milk samples in each study should be hydrolyzed before the analysis so that the results could show the total composition of amino acid in milk, and (3) except for sow milk if any treatments or diets were included, we only select the control group as common practice. Typical information on sows and the methods applied in the present study and six publications is listed in Table 1. The piglets used in the cited literature studies to analyze the amino acid profile of the entire body composition were slaughtered at weaning and had not received any creep feed during the lactation period (Mahan and Shields, 1998). 

### 2.4. Statistical Analysis

The data obtained in this study were expressed as mean ± standard deviation. The level of each amino acid was expressed as the proportion of the whole amino acid concentrations (on a weight basis) both in the current study and the selected publications. The comparison of amino acid content and profile in sow milk between different dates during lactation was conducted in SPSS. 28.0 using a mixed model ANOVA of and post hoc Tukey contrasts to determine the differences. Differences were considered significant at *p <* 0.05. 

## 3. Results

### 3.1. Amino Acid Concentrations in Sow Milk at d 0 (Colostrum), d 3, and d 10 after Parturition

The amino acid content of hydrolyzed sow milk (colostrum) is shown in Table 2. During lactation, the total amino acid content decreased more than 50%, from 946 mmol/L in colostrum (d 0) to 413 mmol/L in milk at d 3, while there was only about a 15% reduction from d 3 to d 10. From d 0 to d 3, each of the individual amino acid concentrations showed significant decreases (*p* < 0.05), while most amino acid concentrations (Thr, Ser, Glu, Gly, Val, Met, Tyr, Phe, His, and Pro) remained stable (*p* > 0.05) from d 3 to d 10. Glutamine + glutamate was the most abundant amino acid, accounting for 137 mmol/L in colostrum, followed by proline, leucine, asparagine + aspartate, and lysine.

### 3.2. Amino Acid Profile of Sow Milk at d 0 (Colostrum), d 3, and d 10 after Parturition

Table 3 represents the amino acid profiles of sow colostrum at d 0 and sow milk at d 3 and d 10 of the lactation period. Since the data was expressed as the proportion of total amino acids as a weight basis, the conservation of amino acid profiles among different dates became apparent. Glutamine + glutamate was the most abundant amino acid in sow milk across the sample dates, occupying about 14–17% of the total amino acid content, followed by proline and leucine. Although all actual amino acid concentrations in sow milk decreased significantly throughout the first day of lactation, the proportions of glutamine + glutamate, proline, lysine, and methionine to the total amino acid rose significantly from d 0 to d 3, while those of threonine, serine, leucine, and tyrosine decreased (*p* < 0.05). From d 3 to d 10, most of the AA remained stable (*p* > 0.05) as a proportion of total AA except for asparagine + aspartate, glutamine + glutamate, alanine, and valine in the course of the ranking based on the six highest amino acid proportions remained similar along the course of lactation. 

### 3.3. Amino Acid Profile of Sow Milk across Studies

The amino acid profiles of sow milk throughout lactation showed a similar pattern (Figure 1), even across studies with various factors shown in Table 1. Glutamine + glutamate was the most abundant amino acid in these studies, accounting for around 17–23%. According to different studies, the proportions of glutamine + glutamate and asparagine + aspartate were more variable than other amino acids. There were some minor numerical differences in the ranking based on amino acid abundance between studies. In all studies, the proportions of glutamine + glutamate, proline, leucine, lysine, and asparagine +aspartate, however, all ranked in the top five.

### 3.4. Amino Acid Profile of Milk across Animal Species

Figure 2 shows the amino acid profiles of sow, cow, goat, and human milk. Among these species, most amino acid proportions were relatively similar. Except for human milk, the proportions of glutamine + glutamate, proline, leucine, valine, lysine, and asparagine + aspartate were still the top six most abundant amino acids, accounting for around 18%, 10%, 9%, 7%, 6%, and 8%, respectively. Glutamine + glutamate was the most abundant and methionine was the scarcest amino acid in milk of all four species. Sow milk had relatively more proline, glycine, valine, and threonine, but less or the least glutamine + glutamate (16.09%), lysine (6.81%), and methionine (1.30%) compared to the other three species. Goat and cow milk had higher proportions of glutamine than sow and human milk, and goat milk had higher lysine than the other species. Arginine was the fifth most abundant amino acid in human milk, and human milk had the highest arginine proportion.

### 3.5. Comparing Amino Acid Profiles of Sow Milk with Piglet Body Composition

The amino acid pattern in the whole body of piglets near weaning age (8.5 kg bodyweight) showed a close resemblance with the average amino acid pattern found in sow milk in the present study (Figure 3). Glutamine + glutamate, proline, and leucine were most abundant while methionine was still the scarcest in the amino acid profile in sow milk and piglet body. Two marked differences were found: proline was higher in sow milk than in piglets, whereas arginine was lower.

## 4. Discussion

The amino acid profiles of milk composition included in these studies presented similarities and also noticeable variations. However, there is a large range of internal and external factors that may theoretically influence milk composition (Table 1) though most studies have focused on the macronutrient concentrations of sow milk. Different technologies of amino acid analysis could play a key role in affecting the results since various levels of sensitivity and the process of sample preparation in different methods could influence the results directly. In the present study, an automatic amino acid analyzer with an ion-exchange chromatography (IEC) method was applied, while others applied precolumn derivatization preceding RP-HPLC. Due to the long run time of IEC, the ninhydrin is unstable and the other analytes may react with the ninhydrin, whereas this method is highly reproducible and has good linearity over a broad range [35,36]. Therefore, the selection of technology could also be the factor causing variance.

As colostrum and milk are the main nutrition sources for neonatal pigs, they can have a profound impact on neonates in the short and long term through physiological processes [37]. The data of this study and the comparison with published studies showed that the amino acid profile (expressed as the proportions of total amino acids) of sow milk is conserved throughout the lactation period and is only marginally influenced by various factors and even different breeds. However, the actual concentrations of amino acids in sow milk decreased significantly during the first days of lactation. As the nutritional component of maternal milk is considered the “gold standard” for infant feeding, WHO21, [38], infant formulae are optimized based on milk composition. Therefore, the results of our study suggest that this conserved amino acid profile of sow milk is suited for the nutritional requirement of piglet development and may be used as a guide for the design of creep feed for preweaning piglets.

In previous studies, the modulation of diet structure and the inclusion of feed additives has been applied to sow diets to optimize the nutritional components (protein, fat, and lactose) of sow milk. Several studies reported that the changes in dietary fat level and source could alter both fat and lactose output, and the fatty acid profile of sow milk [39,40,41]. King et al. [42] found that an increase in the dietary protein level could improve the total protein concentration of sow milk, though the proportions of each amino acid were unaffected. Xu et al. [43] reported that the amino acid concentrations of sow colostrum increased with the standardized ileal digestible valine:lysine ratio, but the proportions of each amino acid to total amino acids were stable. These findings combined with the results of the present study suggest that the macronutrient profile is not as conserved as the amino acid profile in sow milk, suggesting that there is more flexibility in exploring better nutrition sources for sow diets while the prospects of applying specific amino acids to sows will likely not result in changes in the amino acid profile.

In the sow milk samples of our study, the actual amino acid concentrations decreased, though the amino acid pattern remained stable during lactation, in agreement with data reported by Aguinaga et al. [44] and Csapo et al. [24]. Even though there were minor differences among these three studies, the top six most abundant amino acids were the same and the rankings based on amino acid abundance were similar. That was also the case in other studies that had analyzed the amino acid profile of sow milk [44,45]. 

Glutamine plus glutamate were the most abundant amino acids independent of the lactation stage and study, accounting for 17–23% of the total amino acids. This points to the high requirement for piglet development. Glutamine has been proposed as an indispensable amino acid to all tissues, since glutamine, as a protein synthesis precursor, has many metabolites including glutamate, alanine, aspartate, α-ketoglutarate, and arginine that are involved in various metabolic processes [46]. Glutamine, as the preferred fuel for intestinal development and immune cells [13,15], is utilized at approximately 67% by the small intestine of the pig, and it is the only amino acid in arterial blood absorbed by the small intestine [47]. A myriad of studies reported the positive effect of glutamine on piglet-growth performance, intestinal epithelium structure, antioxidant capacity, and barrier functions [48,49,50,51], which are exactly suitable for newborn piglets with immature digestive functions. Likewise, proline and leucine were also abundant amino acids. Proline, as the conditionally essential amino acid, is highly abundant in collagen and elastin [52], and adequate proline is essential for maximal collagen synthesis and growth performance for animals [53,54,55]. Apart from these functions, proline also seems to be a precursor for arginine synthesis in neonates [56]. It is therefore an interesting observation that the higher proline content in sow milk may be a source for the higher arginine content in the piglet body. Dietary amino acids are extensively catabolized by the small intestine in the first pass, and nearly 30–40% of all proline is degraded [57,58]. These reports above also contributed to the discrepancies in the comparison of the present study. However, it is not clear what drives the level of these conversions, though it suggests an underestimated role of proline in suckling piglet nutrition.

Leucine, as a BCAA, is not only the substrate for protein synthesis. It can also stimulate protein deposition in skeletal muscle by serving as a signal molecule or mTOR signaling pathway [59,60]. Leucine also performs a main role in the catabolism within the BCAA, since the complex branched-chain α-keto acid dehydrogenase, which is a key enzyme in BCAA catabolism, is regulated by leucine intake [61].

Among the different species, the amino acid profiles of milk generally matched in the numerical comparison, despite minor discrepancies. The typical most abundant amino acids were similar for these four mammalian species, i.e., glutamine + glutamate, proline, valine, leucine, lysine, and asparagine, accounting for over 60% of total amino acids. It should be noted that the level of “Glu” presented in the results is the sum of glutamine and glutamate, and “Asp” is the sum of asparagine and aspartate due to sample hydrolysis for amino acid analysis. This may lead to an overestimation of their importance, though the above arguments do support their importance. It would, however, be interesting for instance to know the glutamate: glutamine ratios in milk because exactly the interconversion between glutamine and glutamate is under strong feedback regulation of several other amino acids and nutrients [62,63,64].

Sow milk had numerically higher proportions of glycine, valine, and proline than other species. Wang et al. [65] concluded that glycine is a dietary-essential amino acid for maximal protein accretion in milk-fed piglets and that glycine is also the most abundant free amino acid in the plasma of pigs [66]. Valine as a BCAA is crucial for regulating the energy homeostasis, intestinal health, and immunity, which is essential for piglets under intensive production [67]. The amino acid profiles of cow and goat milk were relatively close, and it also was reported that the amino acid sequence of cow and goat milk protein showed an 88% homology [68]. Even though the amino acid sequence of cow and goat milk has been concluded to share a 60% homology with human milk, infant formulae made from these two milks still required modification of individual amino acids due to the higher proportion of methionine and phenylalanine in cow and goat milk [69,70,71,72]. There was a report that an infant fed with undiluted goat milk showed signs of severe liver impairment, which was related to the excessive amounts of methionine and phenylalanine in goat milk [69]. The similar digestive physiology between pigs and humans, and close proportions of methionine and phenylalanine in their milk, as shown in our study, may point to an inaccurate attempt at diet design for piglets as well. Interestingly, this general similarity in amino acid profiles was also observed in nonmammalian body compositions, with the predominant amino acids in frogs (*Xenopus laevis*), such as glutamine, glycine, leucine, and asparagine being generally similar to those in fish species (*Engraulis encrasicholus*), while methionine, histidine, and phenylalanine were scarce in both [73]. By contrast, sows are intensively using methionine as a methyl donor during lactation [74].

Generally, to mimic sow milk, creep feed is often nutrient dense and formulated with more highly digestible ingredients such as milk byproducts to facilitate the nutrient intake of immature piglets’ digestive systems [75]. In addition, the level of protein in creep feed is normally higher than that in sow milk [76]. However, highly digestible creep feed only improves the performance of piglets during the preweaning period, and coarse grain-based creep feed benefits postweaning performance more than highly digestible creep feed [77]. Therefore, attention to the amino acid profile of creep feed in close connection with the weaning diet might be a new impetus for creep feed development.

Finally, the conserved amino acid pattern in milk since d 3 might reflect the nutrition requirement of piglets in the suckling period. However, whether the overall amino acid structure or the partial amino acid combination can meet piglets’ needs efficiently remains a question. Although the nutritional composition of supplementary feed for suckling piglets can be modified based on data on maternal milk composition, the precise level of conversion occurring in the first pass in the intestines needs further exploration. It also remains to be verified whether the conserved amino acid structure is already an ideal state for alleviating stress from diet transition in the future or the restriction to the development of preweaning piglets now.

In conclusion, the amino acid pattern of sow milk is relatively stable across lactation stages and diets. Compared to other species, sow milk is rich in glycine, valine, and proline, the latter most likely serves as a source for arginine synthesis in the piglets. These findings provide a basis to improve the amino acid pattern of creep feed or at least identify the amino acid needs of suckling piglets, even beyond essential amino acids.

## Figures and Tables

**Figure 1 vetsci-10-00298-f001:**
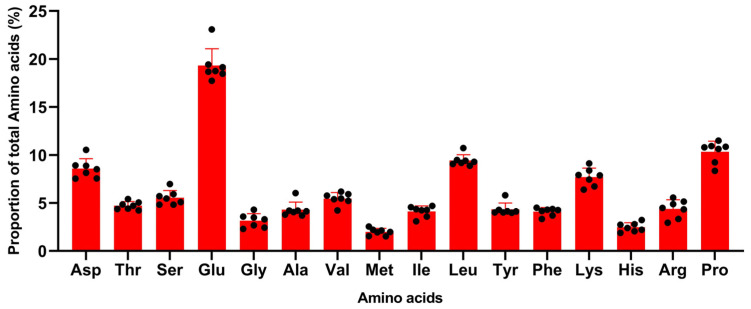
Amino acid profile of sow milk across studies (% of total amino acids, the black spot were the data collected from the studies shown in Table 1) (Wu et al., 1994, Csapó et al., 1996, Kim et al., 2004, Beyer et al., 2007, Mudd et al., 2016, Rezaei et al., 2022).

**Figure 2 vetsci-10-00298-f002:**
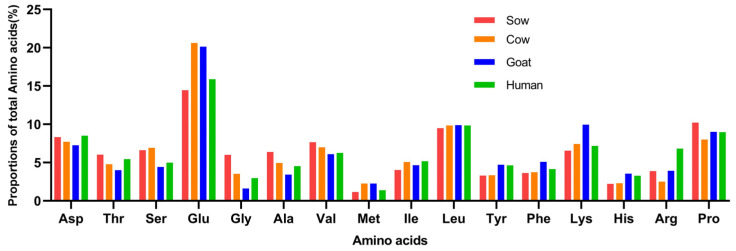
Amino acid profile of milk across animal species (% of total amino acids) (Ceballos et al., 2009, Garcia-Rodenas et al., 2016, Gu et al., 2020).

**Figure 3 vetsci-10-00298-f003:**
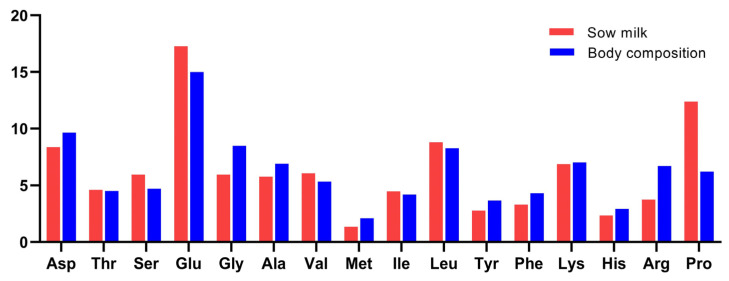
Amino acid profile of sow milk in this study and whole piglet body in 8.5 kg body weight (% of total amino acid).

**Table 1 vetsci-10-00298-t001:** An overview of potential factors influencing sow milk composition in different studies.

	Sow Breed	Number of Sows	Parity (Range)	Sampling Date (Day of Lactation)	Method
This report	Topigs 20	25	3.65 (1–7)	D3	IEC ^a^
Beyer et al. (2007)	Large White × German Landrace	24	2.41 (1–4)	D3	IEC
Wu et al. (1994)	F1 hybrid	10	3.3 (-)	D3	Pre-column with RP-HPLC ^b^
Mudd et al. (2016)	Yorkshire	14	-	D7	IEC
Rezaei et al. (2022)	Yorkshire × Landrace	30	2.5 (2–3)	D3	Pre-column with RP-HPLC ^b^
Csapo et al. (1998)	Danish Large White + Duroc + Landrace	30	-	D5	IEC
Kim et al. (2004)	PIC Cambrough-22	10	-	D7, 14, 21	Pre-column with RP-HPLC ^b^

^a^ IEC: ion-exchange chromatography. ^b^ RP-HPLC: reversed-phase high-performance liquid chromatographic.

**Table 2 vetsci-10-00298-t002:** Actual amino acid content of sow milk at d 0 (colostrum), 3, and 10 of lactation (mmol/L, *n* = 25).

Amino Acids	Date of Lactation			*p* Value
	D 0	D 3	D 10	
Asp	78.9 ± 14.2 ^a^	35.8 ± 4.4 ^b^	30.1 ± 3.4 ^c^	<0.001
Thr	57.1 ± 12.4 ^a^	19.9 ± 3.4 ^b^	16.6 ± 2.3 ^b^	<0.001
Ser	62.1 ± 12.5 ^a^	24.7 ± 3.9 ^b^	21.2 ± 2.9 ^b^	<0.001
Glu	137.0 ± 23.7 ^a^	68.4 ± 7.9 ^b^	61.6 ± 6.6 ^b^	<0.001
Gly	56.0 ± 11.2 ^a^	23.5 ± 2.9 ^b^	20.9 ± 2.9 ^b^	<0.001
Ala	59.8 ± 11.9 ^a^	24.6 ± 3.2 ^b^	20.5 ± 2.6 ^c^	<0.001
Val	72.5 ± 13.7 ^a^	26.2 ± 3.7 ^b^	21.9 ± 3.5 ^b^	<0.001
Met	11.0 ± 2.6 ^a^	5.5 ± 9.9 ^b^	4.9 ± 0.8 ^b^	<0.001
Ileu	38.3 ± 7.2 ^a^	18.5 ± 2.7 ^b^	16.1 ± 2.1 ^c^	<0.001
Leu	90.2 ± 17.8 ^a^	37.5 ± 4.9 ^b^	32.0 ± 4.1 ^c^	<0.001
Tyr	31.3 ± 6.5 ^a^	11.8 ± 2.0 ^b^	10.1 ± 1.5 ^b^	<0.001
Phe	34.5 ± 6.8 ^a^	13.8 ± 1.9 ^b^	12.0 ± 1.6 ^b^	<0.001
Lys	62.4 ± 11.2 ^a^	28.6± 3.1 ^b^	24.5 ± 2.7 ^c^	<0.001
His	21.1 ± 4.4 ^a^	9.4 ± 1.3 ^b^	8.2 ± 1.0 ^b^	<0.001
Arg	36.7 ± 6.9 ^a^	15.6 ± 2.2 ^b^	13.3 ± 1.9 ^c^	<0.001
Pro	97.3 ± 16.4 ^a^	49.3 ± 6.7 ^b^	44.7 ± 5.9 ^b^	<0.001
EAA *	423.7 ± 79.3 ^a^	174.8 ± 22.1 ^b^	149.5 ± 18.3 ^c^	<0.001
NEAA *	522.2 ± 93.7 ^a^	237.9 ± 28.8 ^b^	209.1 ± 22.3 ^c^	<0.001
Total AA	945.9 ± 172.3 ^a^	412.8 ± 50.2 ^b^	358.5 ± 40.0 ^c^	<0.001

Values were expressed as mean ± standard deviation, *n* = 25. Values followed by different superscript ^a–c^ within a row were significantly different (*p* < 0.05). * EAA: essential amino acid; NEAA: nonessential amino acid. Asp and Glu represent Asn + Asp and Glu + Gln, the following tables and figures are the same.

**Table 3 vetsci-10-00298-t003:** Amino acid profile of sow milk at d 0, 3, and 10 of lactation (% of total amino acid, *n* = 25).

Amino Acids	Date of Lactation			*p* Value
	D 0	D 3	D 10	
Asp	8.6 ± 0.2 ^c^	8.9 ± 0.2 ^a^	8.7 ± 0.2 ^b^	<0.001
Thr	5.6 ± 0.5 ^a^	4.4 ± 0.4 ^b^	4.3 ± 0.3 ^b^	<0.001
Ser	5.4 ± 0.2 ^a^	4.8 ± 0.3 ^b^	4.8 ± 0.4 ^b^	<0.001
Glu	16.2 ± 0.5 ^c^	18.5 ± 0.2 ^b^	19.2 ± 0.8 ^a^	<0.001
Gly	3.5 ± 0.3 ^a^	3.3 ± 0.3 ^b^	3.4 ± 0.5 ^ab^	0.005
Ala	4.4 ± 0.3 ^a^	3.1 ± 0.30 ^c^	3.9 ± 0.3 ^b^	<0.001
Val	7.0 ± 0.4 ^a^	5.8 ± 0.4 ^b^	5.6 ± 0.4 ^c^	<0.001
Met	1.4 ± 0.2 ^b^	1.6 ± 0.1 ^a^	1.6 ± 0.2 ^a^	<0.001
Ileu	4.2 ± 0.2 ^b^	4.6 ± 0.3 ^a^	4.6 ± 0.2 ^a^	<0.001
Leu	9.8 ± 0.5 ^a^	9.3 ± 0.4 ^b^	9.1 ± 0.5 ^b^	<0.001
Tyr	4.7 ± 0.2 ^a^	4.3 ± 0.3 ^b^	4.3 ± 0.3 ^b^	<0.001
Phe	4.7 ± 0.2 ^a^	4.3 ± 0.2 ^b^	4.3 ± 0.2 ^b^	<0.001
Lys	7.5 ± 0.3 ^b^	7.9 ± 0.3 ^a^	7.8 ± 0.2 ^a^	<0.001
His	2.7 ± 0.1	2.7 ± 0.2	2.8 ± 0.2	0.301
Arg	5.3 ± 0.2 ^a^	5.1 ± 0.3 ^ab^	5.0 ± 0.2 ^c^	0.010
Pro	9.3 ± 0.6 ^c^	10.6 ± 0.8 ^b^	11.1 ± 0.72 ^a^	<0.001

Values were expressed as mean ± standard deviation. Values followed by different superscript ^a–c^ within a row were significantly different (*p <* 0.05).

## Data Availability

The datasets used and/or analyzed during the current study are available from the corresponding author on reasonable request.

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
