# Peer review of "Peculiarities in the Amino Acid Composition of Sow Colostrum and Milk, and Their Potential Relevance to Piglet Development"

_vetsci, 2023, doi:10.3390/vetsci10040298_

Round 1

Reviewer 1 Report

Line 19, change for:  sow colostrum and milk

Lines 123-144, I my opinion this part of material and method should be removed, it is the bibliographic review work that is done whenever a research work is done

Pag 4 The results are repeated: table 2 to figure 1 and table 3 to figure2. You must choose table or figure to present your results. In any case, in figures lack superscripts on the differences between the types of sow's milk.

Table 2 and 3, the superscripts have to go in decreasing order, the highest value with the superscript a

Line 181, methionine is repeated

Lines 196-233, this part could be considered within the discussion, the authors compare the results obtained with the bibliography on the amino acid composition of sow milk, other species and piglets

Line 236 physiologicalmprocesses

Lines 243-352, this paragraph should be moved to the beginning of the discussion

Reviewer 2 Report

Great information on amino acids components in sows milk. 

1. what tissues of piglets were used to analyze the AA concentrations

2. Minor spelling check 

Reviewer 3 Report

Peculiarities in the amino acid composition of sow colostrum 2 and milk, and their potential relevance to piglet development

This paper compares the amino acid profile of sow colostrum and milk across different days in lactation and in addition, compares the data with other published literature in sows and in other species. This is a somewhat unorthodox presentation in that it combines aspects of an original paper and of a review. I find this an interesting approach, since it brings more value to the presented original data, even if though these may not be that novel.

Specific comments

Simple summary.

Poor English compared to Abstract and rest

L12-17 Background, introductory, not to the point

M&M

L99 …water was available unrestrictedly…

L128 …account for nearly 85 % of…

Results

Heading of table 2: ‘Day of lactation’

Figure 1 is basically a repeat of Table 2 and not needed.

L176 proportion: molar or weight?

L183 Rephrase: From d3 to d10, most of the AA remained stable as a proportion of total AA…

L185 …in the course of…

Figure 2 is basically a repeat of Table 3 and not needed.

Discussion

L236-7 correct spelling

L248 …applied to…

L322-332 poor English please correct
